# Enhancement of the Anti-Inflammatory Effect of Mustard Kimchi on RAW 264.7 Macrophages by the *Lactobacillus plantarum* Fermentation-Mediated Generation of Phenolic Compound Derivatives

**DOI:** 10.3390/foods9020181

**Published:** 2020-02-12

**Authors:** Bao Le, Pham Thi Ngoc Anh, Seung Hwan Yang

**Affiliations:** Department of Biotechnology, Chonnam National University, Yeosu, Chonnam 59626, Korea; 178700@jnu.ac.kr (B.L.); 197621@jnu.ac.kr (P.T.N.A.)

**Keywords:** inflammation, mustard leaf, nitric oxide, phenolic

## Abstract

Mustard leaf kimchi contains numerous functional compounds that have various health benefits. However, the underlying mechanisms of their anti-inflammatory effects are unclear. In this study, changes in the mustard leaf kimchi phenolics profile after fermentation with or without *Lactobacillus plantarum* were determined using liquid chromatography–mass spectrometry/mass spectrometry (LC–MS/MS). To correlate changes in phenolic profiles with anti-inflammatory activities of the fermentation extracts, lipopolysaccharides (LPS)-stimulated RAW 264.7 cells were treated with the extracts. We identified 12 phenolic acids in mustard leaf kimchi fermented with *L. plantarum*. Caffeic acid, chlorogenic acid, epicatechin, and catechin substituted the metabolite abundance. Extracts of mustard leaf kimchi fermented by *L. plantarum* (MLKL) markedly inhibited nitric oxide production by decreasing interleukin 6 (IL-6), tumor necrosis factor-α (TNF-α), inducible nitric oxide synthase (iNOS), and cyclooxygenase 2 (COX2) expression levels in LPS-treated RAW 264.7 cells. Thus, fermentation with *L. plantarum* potentially improves the anti-inflammatory activities of mustard leaf and mustard leaf fermented with this microorganism may serve as a proper diet for the treatment of inflammation.

## 1. Introduction

Inflammation is a protective mechanism of organisms against tissue injury caused by wounding, microbial pathogen infections, and chemical irritation [1]. Instead of anti-inflammatory drugs, traditional fermented foods are being developed as alternative therapeutics and useful diets [2]. Fermented food consumption has been considered to reduce inflammatory bowel diseases, although clinical data to support this have not been reported so far [2].

Mustard leaf (*Brassica juncea*), a cruciferous cormophyte vegetable, is widely used as a food spice and an ingredient in traditional medicines [3]. In Korea, it is used either directly as food or as a major ingredient of kimchi, a traditional fermented vegetable-based food [4]. Kimchi containing mustard leaf has recently attracted much attention as a functional food for health maintenance and disease prevention [5,6]. Mustard leaf acts as an effective anti-inflammatory agent against acute and chronic inflammatory processes by suppressing the mRNA expression of a panel of inflammatory mediators, including TNF-α, IL-6, and IL-1β, in mice [7].

Phenolic compounds are essential constituents of food [8]. Being a major component of plants, these compounds are directly related to the sensory characteristics of plant-based foods, such as flavor, stringency, and color [8]. Phenolic compounds are consumed through plant-based foods are beneficial to human health because of their activities against carcinogenesis and mutagenesis, mainly through antioxidant effects [9]. Quercetin has an important anti-inflammatory activity in microglial cells [10].

In the processes of plant and vegetable fermentation, lactic acid bacteria (LAB) can enrich various phenolic compounds by secretion of numerous enzymes [11]. LAB fermentation play an important role in preserving edible food materials, enhancing desirable flavors, and improving the biochemical profiles of vegetables-based foods resulting in potentially enhancing their biological activities [12]. LAB is the predominating bacteria of cruciferous products as cabbage, mustard leaf, and broccoli. During the fermentation process, LAB change the total polyphenol content, enhance antioxidant capacity, and produce glucosinolates and their break-down products [13]. However, research on mustard leaf kimchi fermentation by lactic acid bacteria is limited.

In this study, we hypothesized that the biotransformation of various phenolic compounds in mustard leaf by LAB-mediated fermentation can improve its benefits for health. The main aim of this study was to investigate the inhibitory effects of mustard kimchi fermented with *Lactobacillus plantarum* against lipopolysaccharides (LPS)-induced inflammatory responses and the underlying mechanisms in mouse macrophage RAW 264.7 cells. To this end, we used liquid chromatography–mass spectrometry/mass spectrometry (LC–MS/MS) for analyzing the phenolic profile changes after fermentation of the mustard leaf by *L. plantarum*.

## 2. Materials and Methods

### 2.1. Inoculation of L. Plantarum

*L. plantarum* FB003 was characterized in our previous study [14]. This strain was cultivated aerobically in the de Man, Rogosa and Sharpe (MRS, Difco, Detroit, MI, USA) medium at 30 °C, after which the culture broth was centrifuged at 1000 × *g* for 4 min and the pellets were washed twice with saline solution. This pellet was then diluted with a saline solution and used as a starter inoculum at a concentration of approximately 1 × 10^8^ CFU/mL.

### 2.2. Preparation of Mustard Leaf Samples

Mustard leaves (*Brassica juncea*) were purchased at a local market in Dolsan (Jeollanamdo, Korea). They were sliced to a thickness of 3 cm and mixed with a 10% NaCl solution for 3 h. After washing with water twice, the mustard leaf pieces were placed inside a basket in a refrigerator for more than 2 h to allow the water to drain. Some leave pieces were stored at −80 °C to be used as raw mustard leaves (RML) for further analysis. The remaining were prepared by adding other minor ingredients, including garlic 1.5%, red pepper powder 1.5%, ginger 0.5%, and mixed seasoning 3%. The final NaCl concentration was 2%. Mustard leaf kimchi fermented by *L. plantarum* (MLKL) was prepared by inoculating with a 10% starter inoculum and the control mustard leaf kimchi (MLK) were not inoculated. The mixture was tightly packed in into glass jars and incubated at 4 °C for 2 months in triplicates.

### 2.3. Determination of the pH, Acidity, and Cell Number of Lactic Acid Bacteria

The blended mustard kimchi samples were filtered, and the pH of the filtrate was determined using a pH meter (FP20, Mettler Toledo, Columbus, OH, USA). Additionally, the acidity of the filtrate was determined by measuring the amount of 0.1 N NaOH required to adjust the pH to 8.3. The total cell number of LAB before and after fermentation was determined by counting the colony forming units (CFU) after incubation on an MRS agar plate. The plates were incubated aerobically at 30 °C for 2–3 days.

### 2.4. Preparation of Extracts

Methanol extracts were prepared by extraction of 20 g of dried powdered samples in 200 mL 80% methanol at 25 °C for 24 h in a shaking incubator (BS-21; Jeio Tech., Daejeon, Korea). Then the samples were centrifuged at 12,000 × *g* for 20 min and then filtered through PTFE 0.20 µm membrane (Millex Samplicity Filter, Merck, Darmstadt Germany). The filtrates were concentrated in a rotary evaporator (EYELA N-1000, Rikakkai Co., Ltd., Tokyo, Japan) at 40 °C. The extracts were then lyophilized before analysis.

### 2.5. Determination of Total Phenolic and Total Flavonoid Contents

The total phenolic content (TPC) of the mustard leaf extracts was performed according to the Folin–Ciocalteu procedure [15]. The mustard leaf extracts (50 μL) was mixed with the Folin–Ciocalteu’s reagent (25 μL) and 20% Na_2_CO_3_ (150 μL). After 15 min of reaction time, the absorbance values were measured at 725 nm using a microplate reader (Asys UVM 340, Biochrom, Cambridge, UK). Chlorogenic acid was used as a standard and the TPC content were expressed as mg of chlorogenic acid equivalents (CAE)/g of the extract.

The total flavonoid contents (TFC) were estimated using the protocol established by Chang et al. [16]. In a microplate, 20 μL of each extract was mixed with 10 μL of 1 M potassium acetate and 10 μL of 10% aluminum chloride. The final volume of the reaction mixture was adjusted to 200 μL using distilled water. Following incubation in dark condition for 30 min, the absorbance at 420 nm was measured using a microplate reader. Quercetin was used as positive control, with the TFC content was presented as mg of quercetin equivalents (QAE)/g of the extract.

### 2.6. Liquid Chromatography–Mass Spectrometry/Mass Spectrometry (LC–MS/MS) Analysis

Mustard leaf extracts were loaded in an ultra-high-performance liquid chromatography (UHPLC) instrument equipped with a tandem MS instrument (Nexera model, Shimadzu, Kyoto, Japan) for the quantitative and qualitative analyses of the phenolic compounds. The liquid chromatograph was equipped with LC–20AD binary pumps (Shimadzu), a DGU-20A5R degasser (Shimadzu), a CTO-20AC column oven (Shimadzu), and a SIL-20AC autosampler (Shimadzu). Chromatographic separation was performed on a UPLC column (Shim-pack XR-ODS II, 75 × 2.0 mm I.D., 3 μm, Shimadzu). The injection volume was 10 μL, the flow rate was 0.9 mL/min, and the temperature was set at 40 °C. The sample was eluted using mobile phase A (0.2 M phosphoric acid) and mobile phase B (100% acetonitrile). The sequence of linear gradient with the following proportions of solvent B: 0–10 min from 0% to 15% of B, 10.1–25 min from 15% to 35% of B, 25.1–40 min from 35% to 90% of B, then gradient was returned to 0% of B solvent to re-equilibrate the column for 15 min. MS detection was performed using a Shimadzu LCMS 8040 model triple quadrupole mass spectrometer (Shimadzu). The 18 standard phenolic compounds were used for analysis, and the standard chromatogram obtained was shown in Figure 1.

### 2.7. Cell Culture, Cytotoxicity Assay, and Nitrite Production

The extracted samples were dissolved in DMSO (final concentration of 0.1% (*v*/*v*)) to final concentrations that were appropriate for the tested dose. The murine macrophage cell line RAW264.7 was cultured in DMEM (Difco, Detroit, MI, USA) supplemented with 10% FBS and penicillin (100 U/mL)/streptomycin (100 μ g/mL) at 37 °C in a 5% CO_2_ humidified incubator. After the RAW264.7 cells reached 80–90% confluency, they were pre-treated with extracts for 2 h, followed by treatment with LPS (1 μg/mL). For control groups, RAW 264.7 cells were untreated and treated with only 1 μg/mL LPS were used as negative and positive controls, respectively. Cytotoxicity was assessed using the EZ-Cytox cell viability assay (DAEILLAB Co, Ltd., Seoul, Republic of Korea), according to the manufacturer’s instructions. Subsequently, the nitrite accumulated in culture medium was estimated using the Griess reagent (0.1% naphthylethylenediamine dihydrochloride and 1% sulfanilamide in 2.5% phosphoric acid), as described by Srisook et al. [17].

### 2.8. Cytokine Assay

Levels of TNF-α and IL-1β were determined using ELISA kits (Abcam, Cambridge, UK), according to the recommended instructions.

### 2.9. Reverse Transcription-Polymerase Chain Reaction (RT-PCR)

After treatment with extracts, total RNA in RAW264.7 cells were extracted using TRIzol reagent (Invitrogen, Carlsbad, CA, USA) to investigate the expressions of inflammatory markers. One microgram of RNA was first reverse-transcribed into cDNA using the SuperScript III Platinum One-Step qRT-PCR Kit (Invitrogen, Paisley, UK). The products were then subjected to quantitative PCR assay using the Power SYBR Green PCR Master Mix (Invitrogen, Paisley, UK) and the following primers: iNOS mRNA, 5′-CACGG ACGAGACGGATAG-3′ and 5′-TGCGACAGCAGGAAGG-3′; COX2 mRNA, 5′-GAATGCTTTGGTCTGGTGCCTG-3′ and 5′-GTCTGCTGGTTTGGAATAGTTGC-3′; and β -actin mRNA, 5′ -CACTGTGCCCATCTACGA-3′ and 5′ -TGA TGTCACGCACGATTT-3′. The relative amounts of the analyzed mRNAs were determined using the comparative (2^−ΔΔCT^) method, as previously described by Livak and Schmittgen [18].

### 2.10. Statistical Analysis

The data were analyzed using the SPSS version 22.0 software (SPSS Inc., Chicago, IL, USA). Mean and standard deviations were calculated from three independent experiments. Differences were estimated using a one-way analysis of variance (ANOVA) followed by Tukey’s HSD test. A *p*-value of less than 0.05 was considered statistically significant.

## 3. Results

### 3.1. Effect of Fermentation on pH, Acidity, Phenolic, and Flavonoid Contents of Mustard Leaf Kimchi

Table 1 shows the pH and total acidity values in raw mustard leaf and mustard leaf fermented with and without *Lactobacillus plantarum* FB003. The lowest pH (4.3) was measured in MLKL, followed by MLK (4.6). Generally, total acidity was inversely proportional to pH. The initial LAB cell numbers were higher in MLKL sample (6.3 log CFU/mL) compared to raw mustard leaf (RML) and mustard leaf fermented without inoculum (MLK) due to *L. plantarum* FB003 inoculation. The population of LAB after fermentation was in agreement with the pH reduction rate.

The total polyphenol content (TPC) and total flavonoid content (TFC) of fermented mustard kimchi in different systems are shown in Table 1. The TPC and TFC in RML extract were 361.2 mg CAE/g extract and 8.0 mg CAE/g extract, respectively. Compared to RML, the contents of total phenolic and flavonoid compounds showed different grades of improvement in fermented mustard leaf. Moreover, MLKL showed a significantly increased phenolic content (*p* < 0.05), compared to that of MLK. We did not observe any statistically significant differences TFC among fermented mustard leaves (MLK and MLKL).

### 3.2. Effect of Fermentation on Phenolic Profiles

We applied LC–MS/MS to determine the phenolic profiles of mustard leaf before and after *L. plantarum* fermentation. Among 18 standard compounds, only 12 compounds were observed, and their concentrations are presented in Figure 1 and Table 2. The phenolic compositions of raw mustard leaf and mustard kimchi seem to be characterized by the presence of chlorogenic acid and caffeic acid as the primary compounds in most cases. The four most abundant phenolic components in MLKL were caffeic acid, chlorogenic acid, epicatechin, and catechin, with contents of 1642 ± 13.2, 896.7 ± 29.1, 544.0 ± 30.9, 460.2 ± 6.6, and 393.7 ± 42.1 μg/g, respectively. Moreover, the contents of caffeic acid, epicatechin gallate, rutin, and naringin n MLKL were significantly higher than those in MLK (*p* < 0.05). Taking into account that these phenolic compounds have a strong anti-inflammatory effect, *L. plantarum* bioconversion might be the reason for the improved the anti-inflammatory property of mustard leaf kimchi.

### 3.3. Effect of Mustard Leaf Kimchi on RAW 264.7 Macrophage Survival and NO Levels

To investigate the mechanism of anti-inflammatory of extracts, we treated LPS-stimulated macrophages with different doses of RML, MLK, and MLKL to assess the cell viability by using Ez-Cytox assay, and the results are shown in Figure 2. The cell survival rates at the various doses of mustard leaf extracts were above 90%. Figure 2C shows that treatment of LPS-stimulated macrophages with 16 µg/mL MLKL resulted in a survival rate of <80%. Interestingly, there was a significant decrease in the cell viability starting from 16 µg/mL of MLKL (*p* < 0.05). Thus, the immunomodulating activities of MLKL below 16 µg/mL were tested using macrophages.

The NO production by macrophages stimulated with RML, MLK, and MLKL is shown in Figure 2D. Compared with the control group cells, LPS-stimulated cells showed significantly increased NO production (*p* < 0.01). LPS-stimulated cells treated with RML, MLK, or MLKL showed opposite results. The reduction of NO secretion levels positively correlated with the RML, MLK, or MLKL concentration (*p* < 0.01). The most potent NO suppressive effect was detected in cells treated with 8 µg/mL MLKL and LPS (84.3%). Our findings indicated that MLKL proved to possess the inhibition of NO production potential in LPS-stimulated cells.

### 3.4. Effect of Mustard Leaf Kimchi on Pro-Inflammatory Cytokines

For the further characterization of the inflammation inhibitory effect of mustard leaf kimchi, we tested levels of pro-inflammatory cytokines, including TNF-α and IL-1β, in LPS-stimulated RAW 264.7 cells. The induction of LPS significantly elevated pro-inflammatory cytokine secretion levels, while the RML, MLK, and MLKL extracts prevented the secretion of inflammation-related cytokines (Figure 3). In LPS-stimulated RAW 264.7 cells treated with 8 μg/mL of RML, MLK, and MLKL extracts, TNF-α production was inhibited by 25.1%, 38.3%, and 64.0% and IL-1β production was decreased by 34.8%, 36.7%, and 56.4%, respectively. The RAW 264.7 cells treated only with LPS group demonstrated significantly enhance TNF-α and IL-1β secretion levels. A significant decrease in TNF-α IL-1β secretion levels were observed in the RAW 264.7 cells treated extracted samples, which was dose-dependent compared with the positive group.

### 3.5. Effect of Mustard Leaf Kimchi on iNOS and COX2 mRNA Expression

Next, we analyzed the expression of iNOS and COX2 genes in RAW 264.7 cells using real-time qPCR. The expression levels of these genes were significantly increased in LPS-treated cells (*p* < 0.01). We also found that RML, MLK, and MLKL extracts at doses of 2–8 μg/mL inhibited the iNOS and COX2 expression in a dose-dependent manner (Figure 4). After treatments with extracted samples, the expression of iNOS and COX2 from 0.51 to 0.93 and from 0.43 to 0.81 (β-actin was used as a housekeeping control). The lowest iNOS expression was observed in MLKL extract at doses of 8 μg/mL. The results suggested that fermented mustard leaf dose-dependently attenuated these downregulates the inflammatory response in LPS-activated RAW 264.7 cells.

## 4. Discussion

In the present study, we elucidated the ability of mustard leaf and mustard leaf kimchi to inhibit the production of inflammatory mediators and enzymatic activation in LPS-stimulated macrophages and the underlying molecular mechanisms. Although mustard leaves have been reported to have a high anti-inflammatory activity [7], the anti-inflammatory activity of the extracts of their fermented products are still unknown. To the best of our knowledge, this is the first report describing the mechanisms underlying the anti-inflammatory effects of mustard leave kimchi fermented with *L. plantarum*.

Jang et al. [19], through a sensory study, demonstrated that the sensory profile of kimchi changes over time during the fermentation process, and that after reaching pH 4.2. These results are consistent with the findings of Lim et al. [20], who reported the optimum ripening pH and acidity of cabbage kimchi to be 4.2–4.5 and 0.6–0.8%, respectively. In the present study, at the end of the fermentation, the pH changes in MLKL treatment were similar to the control MLK. These results suggesting that *L. plantarum* FB003 could not reduce the fermentation period. The LAB count of MLK and MLKL ranged from 6.7 to 7.8 log CFU/mL, values in agreement with those previously reported in previous studies [4,20].

Fermentation via LAB may convert various phenolic compounds in plant materials to more bioactive aglycone forms. Ye et al. [21] reported that LAB fermentation significantly increased the TPC in broccoli puree. Swieca et al. [22] also found an increased enrichment of TPC in different plant materials, including soybean, adzuki, and mung bean. These studies indicated that the phenolic compounds in plant tissues are highly susceptible to lactic acid fermentation processes. A similar case of an increase in flavonoid content was observed in the fruit of *Momordica charantia* [23]. The fact that we did not observe any significant differences in the TFC of fermented mustard leaf samples suggests that *L. plantarum* has a weakened ability to modify flavonoids in mustard leaf kimchi. Moreover, changes in the bioactive component profiles of plant materials occurred upon fermentation with LAB. However, *L. plantarum* fermentation reduced both TPC and TFC of olive and apple [24,25].

According to previous studies, quercetin and chlorogenic acid was found as the principal phenolic acid in mustard leaf [3,26,27]. In the present study, we found that fermentation of mustard leaf decreased its chlorogenic acid levels. These data were attributable to esterase activity, which depends on LAB strain used for fermentation. These results also correlated with increase in caffeic acid levels in fermented mustard kimchi. Bel-Rhlid et al. [28] reported that *Lactobacillus* spp. esterase transforms the chlorogenic acid in green coffee extract to caffeic acid. After fermentation, catechin was identified in fermented mustard leaf samples. The anti-inflammatory activities of these phenolic compounds (caffeic acid, chlorogenic acid, catechin, and epicatechin) have been elucidated in animal studies and clinical trials [29,30,31,32].

Activated macrophages release many kinds of cytokines to enhance the capacity for immune defense, including the production of pro-inflammatory cytokines, such as TNF-α, IL-1β, and IL-6 [33]. Our results demonstrated that RML, MLK, and MLKL stimulated immune responses in macrophages, which can efficiently inhibit TNF-α and IL-1β production in LPS-stimulated cells. Similar inhibitory activities were observed for the *Brassica juncea* (L.) Czern. et Coss. extract, which suppressed TNF-α, IL-6, and IL-1β production [7]. However, MLKL significantly attenuated pro-inflammatory cytokine levels compared to RML and MLK. Other studies demonstrated that fermentation by *Lactobacillus* could enhance anti-inflammatory effect of extracts of plants, such as oyaksungisan, aloe extract, and cactus cladode extract, through inhibiting the production of pro-inflammatory mediators, including NO, PGE2, TNF-α, IL-6, and their synthesizing enzymes iNOS and COX2 [34,35,36]. In order to identify the mechanism responsible for the reduction of NO production by mustard leaves, the inhibition of iNOS and COX-2 is crucial, because they mediate inflammatory events to produce pro-inflammatory cytokines, such as TNF-α and IL-6. The anti-inflammatory effects iNOS and COX-2 were investigated through the determination of the mRNA expression of the genes encoding them. MLKL caused a markedly higher downregulation of these genes than RML and MLK. Taken together, these findings indicate that the anti-inflammatory activities of mustard leaves are significantly increased after fermentation by *L. plantarum*. The increased anti-inflammatory activity of the mustard leaf kimchi compared to that of raw mustard is directly related to the increase in metabolite contents, especially the phenolic content. Besides, the most abundant phenolic compounds in MLKL, caffeic acid and epicatechin, are known to be anti-inflammatory substances.

In conclusion, our study demonstrated that *Lactobacillus plantarum* enhances the anti-inflammatory activity of mustard leaf kimchi. Mustard leaf kimchi fermented using *L. plantarum* (MLKL) inhibits NO production through the suppression of iNOS and COX2 expression in LPS-stimulated RAW264.7 macrophages. MLKL was found to inhibit pro-inflammatory cytokines, including TNF-α and IL-1β. The inflammation inhibitory effects of mustard kimchi fermented using *L. plantarum* described above are attributable to the dephosphorylation of the MAPK signaling pathway. Therefore, we suggest that MLKL can be used as a beneficial food for human health.

## Figures and Tables

**Figure 1 foods-09-00181-f001:**
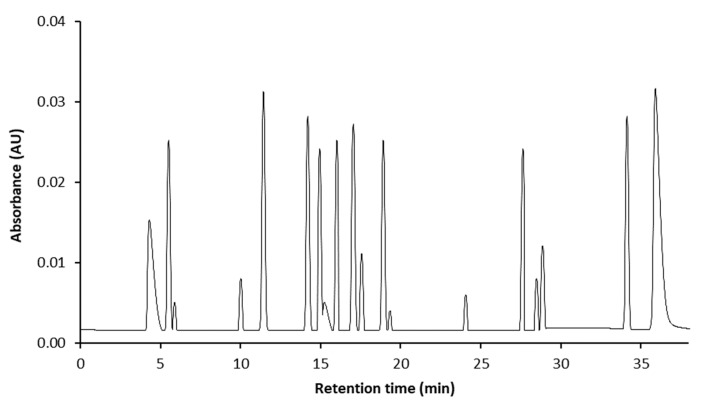
The ultra-high-performance liquid chromatography (UHPLC)–MS/MS chromatograms of the 18 phenolic compounds in standard mixture. 1, catechol; 2, chlorogenic acid; 3, protocatechuic acid; 4, epicatechin; 5, catechin; 6, vanillic acid; 7, caffeic acid; 8, syringic acid; 9, epigallocatechin gallate; 10, *p*-coumaric acid; 11, gallocatechin gallate; 12, epicatechin gallate; 13, rutin; 14, naringin; 15, ferulic acid; 16, sinapic acid; 17, quercetin; 18, kaempferol.

**Figure 2 foods-09-00181-f002:**
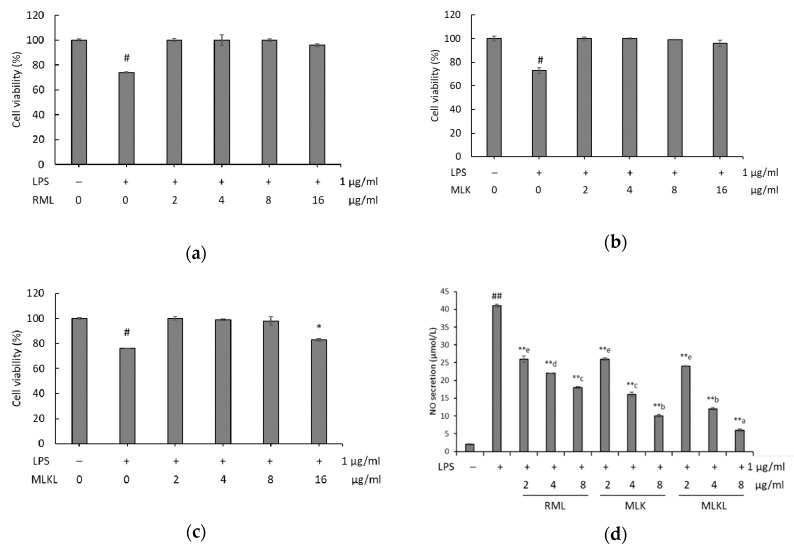
Effect of (**a**) raw mustard leaves (MLK); (**b**) mustard leaf kimchi (MLK); (**c**) mustard leaf kimchi fermented by *L. plantarum* (MLKL) on protective effect and (**d**) NO production in lipopolysaccharides (LPS)-induced RAW264.7 cells. Data were presented as the mean ± standard deviation (SD) of three independent experiments. Means sharing the same alphabet in superscript within each sample are not significantly different at *p* < 0.05 by ANOVA and Duncan’s multiple range test. * *p* < 0.05, ** *p* < 0.01 vs. LPS, and # *p* < 0.05, ## *p* < 0.01 vs. the LPS-stimulated group.

**Figure 3 foods-09-00181-f003:**
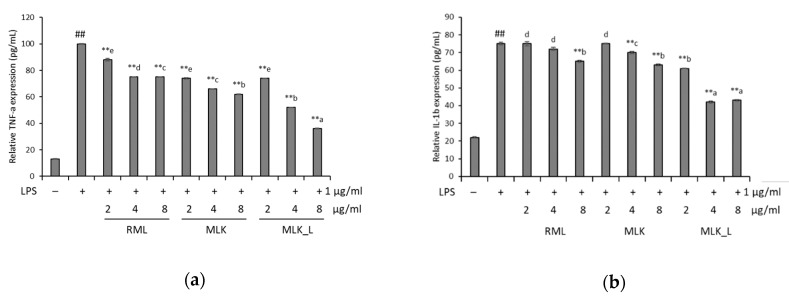
Relative expression levels of tumor necrosis factor (TNF)-α (**a**) and IL-1b (**b**) in LPS-induced RAW264.7 cells. Data were presented as the mean ± standard deviation (SD) of three independent experiments. Means sharing the same alphabet in superscript within each sample are not significantly different at *p* < 0.05 by ANOVA and Duncan’s multiple range test. * *p* < 0.05, ** *p* < 0.01 vs. LPS, and # *p* < 0.05, ## *p* < 0.01 vs. the LPS-stimulated group.

**Figure 4 foods-09-00181-f004:**
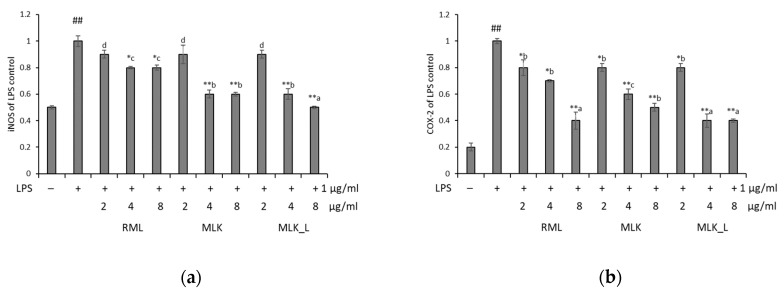
Effect of mustard leaves on the expression of related genes in LPS-induced RAW264.7 cells. The expression of (**a**) iNOS and (**b**) COX-2 by RT-PCR. Data were presented as the mean ± standard deviation (SD) of three independent experiments. Means sharing the same alphabet in superscript within each sample are not significantly different at *p* < 0.05 by ANOVA and Duncan’s multiple range test. * *p* < 0.05, ** *p* < 0.01 vs. LPS, and # *p* < 0.05, ## *p* < 0.01 vs. the LPS-stimulated group.

**Table 1 foods-09-00181-t001:** Total polyphenol and flavonoid contents of mustard leaf kimchi extracts.

Sample	pH	Acidity (%)	LAB Population (log CFU/mL)		Total Polyphenol Content (mg CAE/g Extract Powder) ^1^	Total Flavonoid Content (mg GAE/g Extract Powder)
			Initial	Final		
RML	5.8 ± 0.2 ^b^	0.2 ± 0.06 ^b^	2.7 ± 0.7 ^b^	2.8 ± 0.4 ^c^	361.2 ± 2.1 ^c^	8.0 ± 0.5 ^b^
MLK	4.6 ± 0.6 ^a^	0.7 ± 0.07 ^a^	2.5 ± 0.4 ^b^	6.7 ± 0.5 ^b^	478.6 ± 2.9 ^b^	12.2 ± 0.1 ^a^
MLKL	4.2 ± 0.3 ^a^	0.6 ± 0.04 ^a^	6.3 ± 0.1 ^a^	7.8 ± 0.4 ^a^	512.1 ± 1.8 ^a^	12.9 ± 0.8 ^a^

^1^ Value was presented as mean ± standard deviation (*n* = 3). Means with different letters in the same column are significantly different based on Tukey’s HSD test (*p* < 0.05).

**Table 2 foods-09-00181-t002:** Changes in phenolic concentration in mustard leaf.

No	Analytes	RT ^1^	Extract (μg/g)		
			RML	MLK	MLKL
1	catechol	4.4	- **^2^**	-	-
2	chlorogenic acid	5.3	1630.2 ± 23.1 ^a,A^	861.2B ± 23.5 ^b,B^	896.7 ± 29.1 ^b,B^
3	protocatechuic acid	5.9	-	-	-
4	epicatechin	10.2	263.5 ± 14.2 ^b,C^	467.4 ± 22.0 ^a,C^	460.2 ± 6.6 ^A,c^
5	catechin	11.1	-	324.4 ± 33.3 ^a,D^	393.7 ± 42.1 ^a,C^
6	vanillic acid	14.2	123.4 ± 2.3 ^F^	132.9 ± 3.2 ^F^	111.9 ± 4.7 ^G^
7	caffeic acid	15.1	632.9 ± 21.3 ^c,B^	1213.4 ± 26.4 ^b,A^	1642 ± 13.2 ^a,A^
8	syringic acid	15.3	145.7 ± 6.6 ^E^	111.2 ± 19.4 ^F^	124 ± 43.2 ^G^
9	epigallocatechin gallate	15.8	231.1 ± 32.2 ^a,D^	196.2 ± 24.1 ^a,b,F^	167.9 ± 23.8 ^b,G^
10	*p*-coumaric acid	16.9	150.3 ± 6.4 ^b,E^	231.2 ± 30.1 ^a,F^	246.0 ± 26.4 ^a,F^
11	gallocatechin gallate	17.3	-	-	-
12	epicatechin gallate	18.7	267.6 ± 17.8 ^b,C^	133.8 ± 3.5 ^c,F^	362.6 ± 11.8 ^a,D^
13	rutin	19.1	160.1 ± 42.5 ^b,E^	228.7 ± 19.9 ^b,D^	305.3 ± 18.2 ^a,E^
14	naringin	24.2	532.1 ± 22.5 ^a,B^	296.7 ± 24.1 ^b,D^	144.0 ± 30.9 ^c,G^
15	ferulic acid	27.5	112.3 ± 17.2 ^b,F^	293.4 ± 43.1 ^a,D^	234.6 ± 18.7 ^a,b,F^
16	sinapic acid	28.6	-	-	-
17	quercetin	29.1	-	-	-
18	kaempferol	34.4	351.4 ± 23.7 ^B^	364.6 ± 53.5 ^C^	359.0 ± 12.8 ^D^

^1^ RT, retention time. ^2^ Mean ± standard deviation (*n* = 3). Means with different lower letters in the same row are significantly different based on Tukey’s HSD test (*p* < 0.05). Means with different upper letters in the same column are significantly different based on Tukey’s HSD test (*p* < 0.05). -, not detected.

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
