# Peer review of "Enhancement of the Anti-Inflammatory Effect of Mustard Kimchi on RAW 264.7 Macrophages by the *Lactobacillus plantarum* Fermentation-Mediated Generation of Phenolic Compound Derivatives"

_foods, 2020, doi:10.3390/foods9020181_

Round 1

Reviewer 1 Report

General comments for authors

The manuscript ID foods-709786, titled “Enhancement of Anti-Inflammatory Effect of Mustard Kimchi on RAW 264.7 Macrophages by the Lactobacillus plantarum Fermentation-mediated Generation of Phenolic Compound Derivatives” fits with the scope of the journal because it

provides new interesting information to the advantages of a fermentation via LAB of Mustard Kinchi to improve health effects. Some minor revisions are required before the publication.

Specific comments

Line 32-33: The English form of the sentence should be revised.

Line 71: Please, specify the atmospheric condition used for Lactobacillus plantarum FB003 growth.

Line 75: Please, specify the atmospheric condition used for fermentation Lactobacillus plantarum FB003.

Line 77-81: The authors used the terms “lactic acid bacteria”, but in the previously paragraph it was specified that the inoculum was prepared using the strain Lactobacillus plantarum FB003. Please, clarify if there were used different LAB or just the strain previously cited. To specify the strain name is preferred because a lot of properties are strain specific and do not be extended to other strains belonging to the same species.

Line 81: Please, specify if the MRS agar plate are incubated at the same condition described above to determine the viable bacterial count.

Lines 129-138: Please, repeat the aim of the assay.  

Lines 181-182: Do the authors means “Ez-Cytox assay” with “MTT assay”? Please, clarify.

Lines 226-229: The section “discussion” lacks and should be written.

Lines 238-239: Please, revise the structure of the sentences. Jang et al (2016), through a sensory study, demonstrated that the sensory profile of kimchi changes over time during the fermentation process, and that after reaching pH 4.2 the fermentation level is often regarded as the optimal by many Koreans and not that the pH is equal to 4.2.

Lines 242-243: Please, clarify the sentence “LAB could induce acidity and pH changes to reduce the fermentation period”. Do the fermentation via LAB brought to acidity and pH reduction and thus to the reduction of the fermentation time?

Lines 231-282: The lines from 231 to 282 should be written in the “Discussion” section.

The paragraph 3.4 and 3.5 should be improved, comparing the results obtained from RML, MLK and MLK_L and between tested cytokines and iNOS and COX2 mRNA expression.

Author Response

Author's Reply to the Review Report

The manuscript ID foods-709786, titled “Enhancement of Anti-Inflammatory Effect of Mustard Kimchi on RAW 264.7 Macrophages by the Lactobacillus plantarum Fermentation-mediated Generation of Phenolic Compound Derivatives” fits with the scope of the journal because it provides new interesting information to the advantages of a fermentation via LAB of Mustard Kinchi to improve health effects. Some minor revisions are required before the publication.

Specific comments

Line 32-33: The English form of the sentence should be revised.

Response: Thank you for your comments. We revised the sentence as “Mustard leaf (Brassica juncea), a cruciferous cormophyte vegetable, is widely used as a food spice and an ingredient in traditional medicines”.

Line 71: Please, specify the atmospheric condition used for Lactobacillus plantarum FB003 growth.

Response: Thanks for the referee’s kind advice. We confirmed the atmospheric condition was aerobic. We revised in manuscript as “This strain was cultivated aerobically in the de Man, Rogosa and Sharpe (MRS) medium at 30oC.

Line 75: Please, specify the atmospheric condition used for fermentation Lactobacillus plantarum FB003.

Response: According to reviewer suggestion, we revised the condition.

Line 77-81: The authors used the terms “lactic acid bacteria”, but in the previously paragraph it was specified that the inoculum was prepared using the strain Lactobacillus plantarum FB003. Please, clarify if there were used different LAB or just the strain previously cited. To specify the strain name is preferred because a lot of properties are strain specific and do not be extended to other strains belonging to the same species.

Response: Thank you for pointing this out. To evaluate L. plantarum FB003 as start culture and application in food industry, we didn’t autoclave mustard leaves before fermentation. The fermentation was processed as traditional procedure with supplements of L. plantarum FB003 as inoculum. Therefore, we need to estimate population of all LAB in fermented product, not only L. plantarum FB003 strain. We also revised the sentence in manuscript as “The total cell number of LAB was determined by counting the colony forming units (CFU) after incubation on an MRS agar plate. The plates were incubated aerobically at 30°C for 2–3 days.”

Line 81: Please, specify if the MRS agar plate are incubated at the same condition described above to determine the viable bacterial count.

Response: Thank you. We revised the methods as mentioned above.

Lines 129-138: Please, repeat the aim of the assay.

Response: According to reviewer suggestion, we revised the sentence as “ After treatment with extracts, total RNA in RAW264.7 cells were extracted using TRIzol reagent (Invitrogen, USA) to investigate the expressions of inflammatory markers.”

Lines 181-182: Do the authors means “Ez-Cytox assay” with “MTT assay”? Please, clarify.

Response: Thank you for point this out. We change the assay.

Lines 226-229: The section “discussion” lacks and should be written.

Response: The authors sorry about this wrong format. We edited the format of discussion.

Lines 238-239: Please, revise the structure of the sentences. Jang et al (2016), through a sensory study, demonstrated that the sensory profile of kimchi changes over time during the fermentation process, and that after reaching pH 4.2 the fermentation level is often regarded as the optimal by many Koreans and not that the pH is equal to 4.2.

Response: Thanks for the referee’s kind advice. According to the reviewer’s comments, we revised the sentence.

Lines 242-243: Please, clarify the sentence “LAB could induce acidity and pH changes to reduce the fermentation period”. Do the fermentation via LAB brought to acidity and pH reduction and thus to the reduction of the fermentation time?

Response: Thank you for point this out. We totally agree with reviewer that our results can’t confirmed this conclusion. Therefore, we revised this sentence as “In the present study, at the end of the fermentation, the pH changes in MLKL treatment were similar to the control MLK. These results suggesting that L. plantarum FB003 couldn’t reduce the fermentation period.”

Lines 231-282: The lines from 231 to 282 should be written in the “Discussion” section.

Response: We arranged the Discussion section.

The paragraph 3.4 and 3.5 should be improved, comparing the results obtained from RML, MLK and MLK_L and between tested cytokines and iNOS and COX2 mRNA expression.

Response: Thanks for the referee’s kind advice. We improved the section 3.4 and 3.5.

Reviewer 2 Report

This paper entitled, “Enhancement of Anti-Inflammatory Effect of Mustard Kimchi on RAW 264.7 Macrophages by the Lactobacillus plantarum Fermentation-mediated Generation of Phenolic Compound Derivatives” is interesting, it addresses the protective effects of lactic acid bacteria in fermented food.

The study is of interest as it shows the mechanisms underlying the anti-inflammatory effect of Kimchi supplemented with a Lactobacillus plantarum strain. The authors investigated Lactobacillus-mediated protective effects in vitro using macrophages.

However, some questions are raised and need to be addressed by the authors:

The kimchi is a traditional fermented product, it is known that lactic acid bacteria (LAB) including different species of Lactobacillus, represent the core flora responsible for the fermentation. It is not clear why the authors are adding another strain of Lactobacillus at the end of fermentation.

The study design is not clear, and the authors did not have controls to prove that the effect is due to L. plantarum exclusively and is not accumulative.

No bacterial viability data are available. The authors need to include total Lactobacillus counts in mustard leaf kimchi before L. plantarum inoculation (baseline) and after 3 days of inoculation (in MKLM) to show that the anti-inflammatory effects correlate with the bacterial counts. 

Other minors corrections to be implemented:  

Line 32: Originated from China, please correct

Line 46: play an important role

Line 48: The sentence is not clear, please correct for: LAB are the predominating bacteria of cruciferous,

Line 49: Lab change (deleted S). Please note that LAB (lactic acid bacteria) are plural

Line 46: please correct for stored as such at -80

Line 68: Usually the kimchi fermentation occurs during 3 months, why did the author choose two 2 months?

Line 80-81: It is not clear at what time the LAB were counted

117-124: it is not clear which control do the authors use for this study

In results section: Table 1: doesn’t show a significant difference between the pH of MLK and MLKL, which proves that Lactobacillus did not ferment the MLK. It is not clear here how the total phenolic content was increased after MLKL inoculation, while no difference was observed in flavonoid between the groups.

Author Response

Author's Reply to the Review Report

This paper entitled, “Enhancement of Anti-Inflammatory Effect of Mustard Kimchi on RAW 264.7 Macrophages by the Lactobacillus plantarum Fermentation-mediated Generation of Phenolic Compound Derivatives” is interesting, it addresses the protective effects of lactic acid bacteria in fermented food.

The study is of interest as it shows the mechanisms underlying the anti-inflammatory effect of Kimchi supplemented with a Lactobacillus plantarum strain. The authors investigated Lactobacillus-mediated protective effects in vitro using macrophages.

Response: Thank you for careful reviewing our study. The author has appreciated and carefully checked all comments of reviewer. These suggestions from reviewer has helped us to improve our research a lot.

The authors All reviewer commented was

However, some questions are raised and need to be addressed by the authors:

The kimchi is a traditional fermented product, it is known that lactic acid bacteria (LAB) including different species of Lactobacillus, represent the core flora responsible for the fermentation. It is not clear why the authors are adding another strain of Lactobacillus at the end of fermentation.

Response: Thanks for the referee’s kind advice. We agreed completely with reviewer that lactic acid bacteria (LAB) including different species of Lactobacillus, represent the core flora responsible for the fermentation, specially kimchi products. However, due to traditional fermentation process, the concentration of LAB was depending on nutrient competition, starter culture and environment parameters. If LAB will quickly dominate the initial fermentation, the competing microorganisms will not survive, and the end result will be a stable. Lactic acid bacteria (LAB) are widely used as starter cultures in vegetable fermentation. In this study, our purpose is using L. plantarum FB003 as a starter cultures to evaluate the anti-inflammatory effect of mustard kimchi.

The study design is not clear, and the authors did not have controls to prove that the effect is due to L. plantarum exclusively and is not accumulative.

Response: In present study, we used 3 mustard leaf samples including raw mustard leaf (without fermentation), mustard leaf kimchi (with fermentation), and mustard leaf kimchi fermented by L. plantarum  (with fermentation and supplement of L. plantarum (1 × 108 CFU/mL) at beginning). The methods was unclear so we revised the section 2.2.. Please check again in the revision.

No bacterial viability data are available. The authors need to include total Lactobacillus counts in mustard leaf kimchi before L. plantarum inoculation (baseline) and after 3 days of inoculation (in MKLM) to show that the anti-inflammatory effects correlate with the bacterial counts. 

Response:  Thank you for pointing this out. The initial LAB cell numbers were higher in MLKL sample (6.3 log CFU/mL) compared to raw mustard leaf and mustard leaf fermented without inoculum due to L. plantarum FB003 inoculation. The population of LAB after fermentation was in agreement with pH reduction rate. The LAB count of MLK and MLKL ranged from 6.7 to 7.1 log CFU/mL, values in agreement with those previously reported in previous studies (Lim, 2008; Choi, 2001).

Other minors corrections to be implemented:  

Line 32: Originated from China, please correct

Response: We checked this point.

Line 46: play an important role

Response: We changed this mistake.

Line 48: The sentence is not clear, please correct for: LAB are the predominating bacteria of cruciferous,

Response: We changed this mistake.

Line 49: Lab change (deleted S). Please note that LAB (lactic acid bacteria) are plural

Response: We changed this mistake.

Line 46: please correct for stored as such at -80

Response: We changed this mistake.

Line 68: Usually the kimchi fermentation occurs during 3 months, why did the author choose two 2 months?

Response: Thank you for kindly advise. The fermentation process was decided base on traditional process and have been referenced in previous studies on mustard kimchi (Lim, 2008; Choi, 2001; Lee, 2017, Lim, 2000).

Line 80-81: It is not clear at what time the LAB were counted.

Response: The total cell number of LAB before and after fermentation was determined by counting the colony forming units (CFU) after incubation on an MRS agar plate. The plates were incubated aerobically at 30°C for 2–3 days.

117-124: it is not clear which control do the authors use for this study

Response: For control groups, RAW 264.7 cells were untreated and treated with only 1 g/ml LPS were used as negative and positive controls, respectively.

In results section: Table 1: doesn’t show a significant difference between the pH of MLK and MLKL, which proves that Lactobacillus did not ferment the MLK. It is not clear here how the total phenolic content was increased after MLKL inoculation, while no difference was observed in flavonoid between the groups.
Response:  Thank you for pointing this out. To evaluate L. plantarum FB003 as start culture and application in food industry, we didn’t autoclave mustard leaves before fermentation. The fermentation was processed as traditional procedure with or without supplements of L. plantarum FB003 as inoculum. Therefore, no significant difference between the pH of MLK and MLKL was predictable. This result was agreement with previous studies on LAB fermentation. In fact, the further studies are required more detail to estimate microbiome composition of fermented products and enzyme production of microorganisms involved in fermentation.

Reviewer 3 Report

From the extracts of mustard leaf kimchi fermented with Lactobacillus plantarum, these investigators identified 12 phenolic acids, and showed that extracts to exhibit anti-inflammatory activities and could decrease the expression levels of cytokines, NO, iNOS and COX-2. To demonstrate the inflammatory responses, a macrophage RAW cell line was used and LPS was used to activate the inflammation. In addition, chemical assays were used to determine the total polyphenol and flavonoid contents. The ability to profile phenolic components in mustard leaf with possibility to enhance health use is of a good cause. The procedure for quantification of these phenolic compounds is well described. However, the use of a macrophage cell model to test effects of these compounds is not well justified. In addition, there are several problems with the biochemical protocols which are not described in detail, and in particular, the dose of LPS used in this study seems fairly high (1 ug/mL).

Other comments:

More details are needed on the LPS treatments, e.g., incubation time and cell morphology changes should be provided. In the method, how did the authors dissolve the compound? Was DMSO added to the controls? Effects of polyphenols (e.g. quercetin) to attenuate pro-inflammatory cytokine levels should be discussed. Several important papers should be cited in the discussion part. For example, PLoS ONE 10(10): e0141509 and J. Mol. Sci. 2019, 20(4), 932. In all figures, it is not clear what the “a, b, c and d” represent and what are the comparisons. For example, in fig 2d, “b” was observed in MLK and MLKL. Are these two “b”s not different? In fig 3a, there are “b”s in control and in MLK and MLKL, please clarify the differences. Similar problem in fig 3b and 4. Figure 3 legend “Protein production” should be changed to “Relative expression levels”. Line 231 to 282 should be moved to “Discussion” section. Many of the writings need to be improved and clarified. Please check all the acronyms and at least for the first time insert their full name in the abstract (e.g. IL-6, TNF-α, iNOS, and COX2).

Author Response

From the extracts of mustard leaf kimchi fermented with Lactobacillus plantarum, these investigators identified 12 phenolic acids, and showed that extracts to exhibit anti-inflammatory activities and could decrease the expression levels of cytokines, NO, iNOS and COX-2. To demonstrate the inflammatory responses, a macrophage RAW cell line was used and LPS was used to activate the inflammation. In addition, chemical assays were used to determine the total polyphenol and flavonoid contents. The ability to profile phenolic components in mustard leaf with possibility to enhance health use is of a good cause. The procedure for quantification of these phenolic compounds is well described. However, the use of a macrophage cell model to test effects of these compounds is not well justified. In addition, there are several problems with the biochemical protocols which are not described in detail, and in particular, the dose of LPS used in this study seems fairly high (1 ug/mL).

Response: Thank you for kindly advise of reviewer. The authors were considered about this point. We checked the reference of anti-inflammatory activity. The macrophage cell RAW 264.7 cells was model cells for anti-inflammatory in several previous studies (Cheng, 2019; Diao, 2019; Gong, 2020). Moreover, this concentration of LPS was used from 1-2 ug/mL in these studies.

Cheng, R., Wang, L., Li, J., Fu, R., Wang, S., & Zhang, J. (2019). In vitro and in vivo anti‐inflammatory activity of a succinoglycan Riclin from Agrobacterium sp. ZCC3656. Journal of applied microbiology127(6), 1716-1726.

Diao, J., Chi, Z., Guo, Z., & Zhang, L. (2019). Mung Bean Protein Hydrolysate Modulates the Immune Response Through NF‐κB Pathway in Lipopolysaccharide‐Stimulated RAW 264.7 Macrophages. Journal of food science84(9), 2652-2657.

Gong, G., Xie, F., Zheng, Y., Hu, W., Qi, B., He, H., ... & Tsim, K. W. (2020). The effect of methanol extract from Saussurea involucrata in the lipopolysaccharide-stimulated inflammation in cultured RAW 264.7 cells. Journal of Ethnopharmacology, 251, 112532.

Other comments:

More details are needed on the LPS treatments, e.g., incubation time and cell morphology changes should be provided. In the method, how did the authors dissolve the compound? Was DMSO added to the controls?

Response: Thank you for pointing this out. We added more detail in the methods. The extracted samples were dissolved in DMSO (final concentration of 0.1% (v/v)) to final concentrations that were appropriate for the tested dose. For control groups, RAW 264.7 cells were untreated and treated with only 1  mg/ml LPS were used as negative and positive controls, respectively.

Effects of polyphenols (e.g. quercetin) to attenuate pro-inflammatory cytokine levels should be discussed.

Response: Thank you for reviewer suggestion. The authors agreed that quercetin possessed synergistic anti‐inflammatory effects, which may be attributed to their roles in suppressing the activation of TLR4–MyD88‐mediated NF‐κB and mitogen‐activated protein kinases signaling pathways. However,  in present study, the concentration of quercetin was not detected in our samples.

Several important papers should be cited in the discussion part. For example, PLoS ONE 10(10): e0141509 and Int J Mol Sci. 2019 Feb 21;20(4).

Response: Thank you reviewer suggestion. We added these important papers.

In all figures, it is not clear what the “a, b, c and d” represent and what are the comparisons. For example, in fig 2d, “b” was observed in MLK and MLKL. Are these two “b”s not different? In fig 3a, there are “b”s in control and in MLK and MLKL, please clarify the differences. Similar problem in fig 3b and 4.

Response: Thank you for kindly advise. We added in legend figure to explain superscript label as “Means sharing the same alphabet in superscript within each sample are not significantly different at p < 0.05 by ANOVA and Duncan's multiple range test. *p < 0.05, **p < 0.01 vs. LPS, and #p < 0.05, ##p < 0.01 vs. the LPS-stimulated group.”

Figure 3 legend “Protein production” should be changed to “Relative expression levels”.

Line 231 to 282 should be moved to “Discussion” section.

Response: We changed the Discussion section.

Many of the writings need to be improved and clarified. Please check all the acronyms and at least for the first time insert their full name in the abstract (e.g. IL-6, TNF-α, iNOS, and COX2).

Response: We added the full name of all abbreviations.

Round 2

Reviewer 2 Report

All the comments were addressed.

The manuscript can be accepted as it is.